# Stakeholders' perspectives on Public Health Medicine in South Africa

**Virginia E. M. Zweigenthal**[ID]*, **William M. Pick, Leslie London**

School of Public Health and Family Medicine, University of Cape Town, Cape Town, Western Cape, South Africa

* Virginia.zweigenthal@uct.ac.za

## Abstract

South Africa (SA) is reforming its health system in preparation for an anticipated national health insurance (NHI) scheme that aims to improve the delivery of affordable, equitable, accessible health care. Public health (PH) language is explicit in the policy and skilled PH professionals would be expected to play a key role in its implementation. In South Africa, training of doctors as Public Health Medicine (PHM) specialists is funded by the state, yet there are few positions for PHM specialists in the health services. We explored stakeholders' perspectives about this absence, and their views on PHM specialist' roles and contribution in an era of health reform. A qualitative study was conducted in 2012–13, using in-depth interviews with thematic analysis, which elicited perspectives of 31 key stakeholders nationally reflecting diverse employer and institutional backgrounds. While some were surprised by the absence of PH professionals in SA's health system, most agreed the reason was due to factors internal to the profession, such as its low profile and uncertain identity. External factors such as legislation and political preferences for health managers impacted on the employment of PH professionals. However, given the competencies required to implement an ambitious restructuring of the health sector, all believed that PH and PHM personnel were vital. In view of the health system's dominant curative orientation, embedding PH personnel in the services should ensure that health protection, promotion and prevention strategies will inform health priorities. This study, the first known from a low and middle-income country, contributes to the international literature about the identity and roles of PHM physicians, who are versatile professionals with broad skills-sets. In SA, through consultation with health sector employers about potential roles, curricular redesign and trainee recruitment, PHM can graduate fit-for-purpose specialists to work in a range of institutions to address health system reform.

## Introduction

As in many low and middle income countries (LMICs), South Africa's (SA's) health system comprises both private–for those with medical insurance–and public services for 84% of her 57.5 million people with no health insurance in 2018.[1] Most health professionals, particularly medical specialists, work in SA's private sector which disproportionately consumes 44% of health expenditure, but only covers about 16% of the population.[1] Public sector health

**Data Availability Statement:** Data cannot be shared publicly because this is a qualitative study and informants were assured of their anonymity and confidentiality. The Ethics Committee approved the study subject to the data being kept

anonymously. Data requests can be made to Prof Marc Blockman, chairperson of the Human Research Ethics Committee, at the University of Cape Town, marc.blockman@uct.ac.za.

**Funding:** The research was conducted with the support of the Harry Crossley Foundation (http://www.thecrossleyfoundation.co.za/) and the National Research Foundation (NRF) of South Africa, (https://www.nrf.ac.za/) to VEMZ.

**Competing interests:** The authors have declared that no competing interests exist.

services are based on 52 districts delivering services, governed by district offices, under nine provincial health departments, who in addition provide hospital services, with the national department largely responsible for health policy. In larger cities, public health (PH) functions, such as environmental health, are delivered by municipal health services, who in addition may provide health care such as immunisation, TB, contraceptive and HIV services. These municipal functions are based on a tradition of providing preventive personal care that predates SA's 1994 democratic transition, but which were codified under the 2003 Health Act,[2] which established the district health system to deliver accessible, equitable health services.

While government funds the training of medical specialists in PH, they are not institutionalised in this district health system. Public Health Medicine (PHM) is a speciality registerable under the Health Professions Council of South Africa (HPCSA), the national regulatory authority for health professionals. Previous research demonstrated that physicians are largely drawn to specialising in PHM by their commitment to social justice and desire to transform the health system, so that public sector health services address the health needs of the public and communities.[3]

Specialist registration in PHM in SA is a requirement for a small number of 'joint appointment' positions in the public sector that have dual university and service responsibilities. Joint posts exist in provinces where nine universities accredited to train PHM specialists are located. In addition, two provinces have a handful of positions exclusively in the public service specifically earmarked for PHM specialists.[4, 5] Otherwise, specialists find employment as Chief Executive Officers (CEOs) or medical managers in hospitals; as senior managers in provincial departments of health; as researchers in universities or research institutions; in NGOs; and as independent consultants.[6]

PH approaches and expertise are evident in current SA policy documents: the 2011–2017 health workforce policy,[7] the National Public Health Institute of South Africa (NAPHISA),[8] and SA's commitment to universal health coverage through the planned National Health Insurance (NHI) scheme.[9] In view of these articulated commitments and planned health system reform in SA, it is unclear why PH personnel are not more embedded in SA's health system.

We hypothesized that the reasons for the paucity of health career opportunities in the public sector may be the result of three perspectives amongst senior managers: (1) PH skills and population health in general may not be prioritised if clinical and managerial skills are seen as key to the success of health reform–i.e. the clinical paradigm is dominant; (2) whilst appreciating PH skills, policy advisors and employers do not appreciate the value of PHM specialists for three reasons a) being poorly informed about the discipline and/or b) believe other staff can do the same work more cheaply i.e. a demand failure and/or c) do not believe PHM graduates are service ready–i.e. not fit-for-purpose; (3) While being perceived favourably, PHM specialists are not attracted to positions as their career intentions lie outside the public sector, so there is no incentive to create such positions–i.e. a supply failure.

We explored key informants' understanding of the roles and potential contribution of PHM specialists in the health sector in SA. Specifically, we examined their perceptions and experiences of PHM's role in the past and present, including reasons for the paucity of PHM service posts, their understanding of public health skills needed in SA, their views about skills-sets of PH professionals in relation to current health reform policies and possible roles for PH professionals and PHM specialists.

## Methods

Qualitative in-depth interviews were conducted with 31 key informants who were current or past senior leaders in the health sector in SA. The study did not intend to be a representative

survey of all stakeholders' opinions but aimed to elicit their insights and perspectives. Interviews, using open-ended questions, are suited to explore opinions and perceptions, and allowed for exploration of factors impacting on PHM, its underlying evolutionary path and future.

All interviews were conducted by the first author (VEMZ) who, a PHM specialist, was conscious of being a stakeholder, and approached the research reflexively in interviews and analysis. She recognised that her prior training, employment and observation of other specialists, informed the issues probed. Reflexivity, acknowledging and being cognisant that personal history influences the research process, can yield more 'valid' research.[10] Using 'bracketing, she set "aside but did not abandon knowledge and assumptions" to attend to "participants' accounts with an open mind. . . whilst being honest and vigilant" which increased research rigour.[11]

Reflexivity in data collection involved selecting informants carefully, listening to responses and probing.[12] In the analysis, reflexivity involved "careful interpretation and reflection", [13] and transcripts were checked for errors, reread and coded and later reread to ensure that data were not left out and there was no drift in code definitions. The meaning and implications of emerging themes were then reflected upon and articulated. We report negative and discrepant information that did not neatly fall into the themes. These processes promote the trustworthiness of the findings, and conform to measures ensuring the research's reliability.[14]

## Sampling

Information-rich informants with historical perspectives were selected for interviews on the basis of occupying various leadership positions in the health sector. Many worked for health services at national, provincial or local government levels in strategic functions, during and since the democratic transition in 1994, and in some cases before. They were drawn from four groups: (1) policy advisors, (2) employers of PHM specialists, (3) trainers of PHM specialists, and (4) other. Policy advisors were consultants to government in advisory/policy making capacities; employers were managers in NGOs, the public and private health sector; trainers were senior staff, present and past, from university PH departments; and 'other' were either PH trained informants employed by non-health senior government or clinicians.

Sampling was purposive and we invited known senior members of training, NGO and research institutions, and the public sector, across a range of geographic contexts with experience of PH or PHM, to participate. 'Snow-ball' sampling–informants suggesting others with contributory insights–ensured that a wide range of views were collected.

S1 Table details the 31 informants interviewed between November 2012 and May 2013. Most had worked as academics or public sector health managers. Eight (26%) were women. Nine worked in national positions and 22, in four provinces: eight in the Western Cape; seven in KwaZulu-Natal; six in Gauteng and one in North-West. All six retirees were academics, and four had taught PHM specialists. A high proportion (71%) of informants were PHM specialists, 27% had other PH training and 14% had none. Most educators (75%) were PHM specialists. The two from the private sector occupied senior executive positions and previously had longstanding senior leadership posts in the SA public sector health service.

## Ethics and logistics

The study complied with the Declaration of Helsinki (Fortaleza, Brazil 2013). Ethics approval for the study was obtained from the Human Research Ethics Committee of the University of Cape Town (HREC 229/2011).

All informants were invited to participate in the study through email invitations, and were interviewed at time and venue of their choice. Four were telephonic interviews and 27 were conducted face-to-face. After reading the information sheet and asking questions, participants

in face-to-face interviews read and signed consent, including for audio-recording. Those telephonically interviewed gave verbal permission for recording. Interviews were transcribed by a professional who was unaware of interviewees' identities. Face-to-face interviews took between 60 and 90 minutes and telephonic ones lasted between 30 and 60 minutes. The interview guide is included with supporting information.

### Data analysis

The accuracy of transcribed interviews was assured by the researcher listening to them whilst reading the transcripts, correcting transcribing errors, and all personal identifiers were removed. Revised anonymised versions were analysed using Atlas.ti (version 7.0.92), and inductive coding, identifying and analysing themes within and across transcripts was conducted. Thematic analysis is a flexible analytical tool which can provide a rich, detailed yet complex account of data.[15] Our emerging 'mind-map' of themes was checked against the research questions, to ensure they were addressed. Quotations illustrating themes are included in the text. Informants are identified by their current role in organisations and numbered–academics (Acad1-6), retired academics (RetAc1-4), health managers (national (NatMan1-2), provincial (ProvMan1-2), district (DistMan1)), policy advisors'/government consultants, including two retired PHM academics (PolAdv1-7), NGOs (NGO1), the private sector (Priv1-2), national institutes (NatIns1-2), and others (Other1-4).

## Results

Despite noting that PHM specialists are largely absent in the services, all informants recognised that the skills sets central to the speciality are important for the success of SA's health care reform and for health services management. Many mentioned 'public health intelligence', the use of information derived from service data and from published and grey literature as valuable for health services planning and implementation. This meta-function is based on working information and monitoring and evaluation systems and requires skills in evidence-based health care, epidemiology, health policy and understanding health systems–all domains core to PHM and PH:

> . . . using the data that is available, that is collected and that just doesn't go anywhere, . . . And then once you've got useful data, to be able to prioritise and to say, these are the needs . . . and identify where there are weaknesses, whether it's in maternal or child or perinatal mortalities, or whatever. (Acad5)

In light of informants' agreement about the importance of PH functions, reasons for the absence of PHM posts were probed. In addition to reasons for PHM specialists' absence in the services, themes emerging from interviews clustered into five areas: current needs of the services; PHM and PH roles in the services; PHM as a speciality and its identity; academic PHM and PHM's future location. These are presented below.

### Reasons for Public Health Medicine's absence

Some were surprised by PH and the PHM speciality's poor prominence in health service planning and implementation:

> We were naively assuming from an idealistic perspective that public health will be the new forté, the new mantra, the new paradigm–and it didn't happen in the eighteen years. (PolAdv5)

As depicted in Fig 1, informants' reasons explaining the absence of PH trained personnel were: (1) the legacy of the health system in SA and PH's place within that; (2) the recent era of the 'generic health manager'; (3) limited human resources; (4) the public sector work environment; (5) the poor profile of the speciality and its broad identity and (6) uneven training programmes.

**The legacy of health services.** Many informants recognised that PH professionals and PHM specialists were involved in both designing and managing health services over the transition post-apartheid to the new democratic system in the 1990s. These professionals and specialists helped draft frameworks for the new health system–the Bill of Rights in the Constitution and the White Paper for the Transformation of the Health Service that led to the 2003 National Health Act. Some PHM specialists became managers in national, provincial government and municipal health services:

> Some of those voices [public health professionals and PHM specialists] were very important to give credibility and scientific rationale to the broad thrust of public policy. (Priv2)

> When the transition happened . . . we [public health professionals and PHM specialists] all had jobs there . . . All [are] people who ended up in influential positions managing the immediate post-apartheid period. (PolAdv3)

Despite these contributions, designated posts such as Medical Officers of Health that required PH qualifications, disappeared from the health services. Reasons given included the health sector's curative orientation, politicians' roles and PHM specialists' demographics.

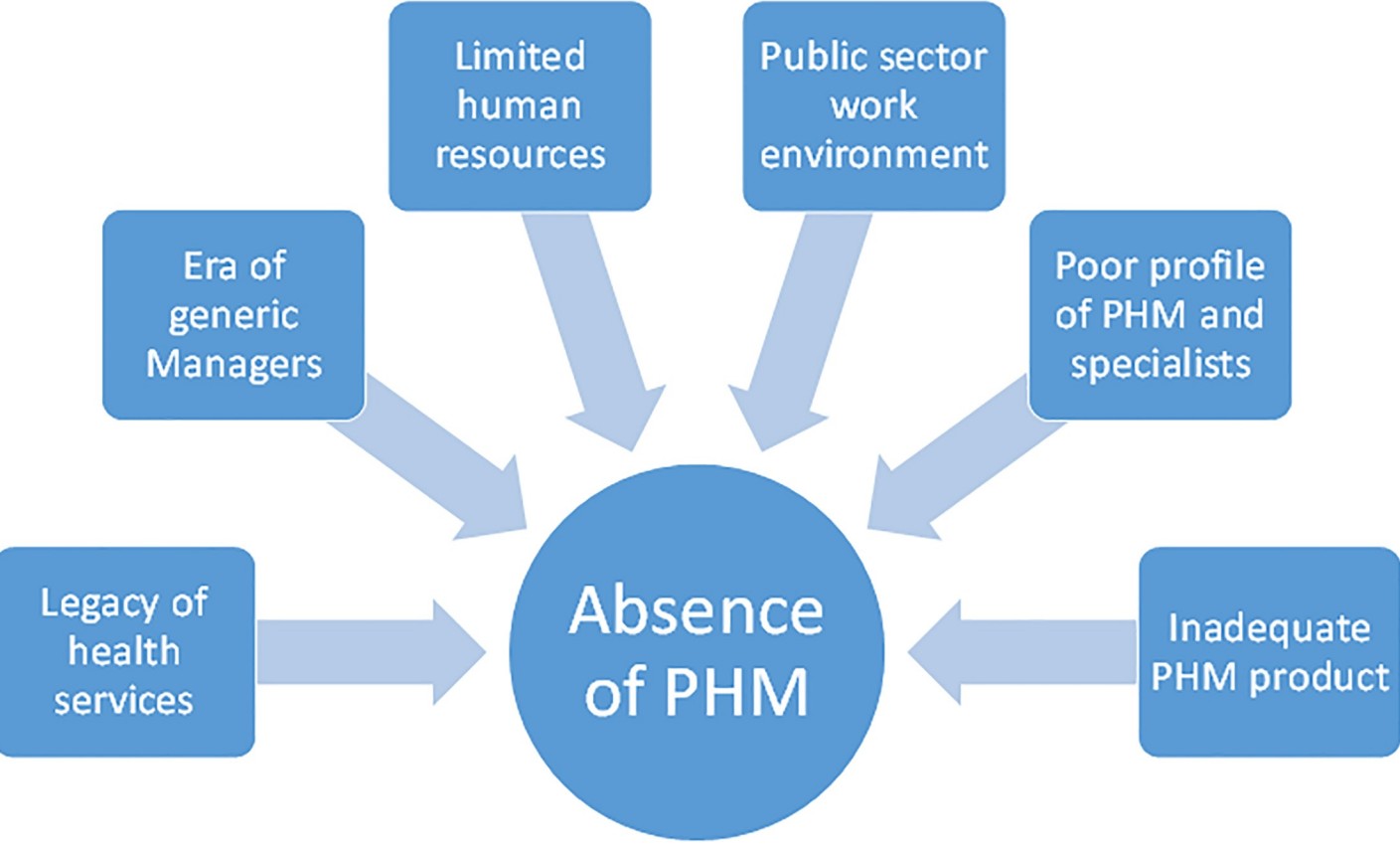

**Fig 1. Reasons for Public Health Medicine's absence in the health services.**

### The health sector's curative orientation

After apartheid, priorities were to finance, enhance and make curatively orientated health services accessible and functional, at the expense of PH imperatives. In addition, the dominant clinical paradigm of SA medicine overshadowed population-oriented approaches:

> One of the most difficult things is to take away from something and put it somewhere else. . . If government in the early days chose to put their new money into public health imperatives and left the funding levels for curative care. . . (NatIns2)

> [When] you are not dealing with emergencies, then public health starts to settle in. But when your work entails covering and plugging and just providing clinical care, there isn't much time to think about public health. (Other3)

> Here [in South Africa] we have an individualist way of doing things. [Medicine] is the stuff of heroes . . . reinforced by hubris. It's not very hubristic to claim, "Oh we helped to prevent so much disease in the communities". It is much more upfront to claim that you did the first heart transplant. (Acad3)

### Politicians' roles

They also believed that preferences of powerful politicians had shaped policies and budgets. Examples given that negatively impacted on PH approaches, were the dismantling of community health worker programmes under the first minister of health, Dr Nkosazana Zuma (1994–1999) and the anti-science perspective of her successor, Dr Manto Tshabalala-Msimang (1999–2008). Consequently, PH approaches, specifically evidence-based practice, hardly shaped service priorities. In addition, senior level appointments were based on cronyism–loyal party members delivering politicians' priorities:

> During Manto's leadership in the Department of Health, there was ambivalence towards scientific medicine . . . and particularly the public health paradigm–and the consequence . . . was a loss of focus on evidence-based medicine and particularly public health approaches to health. (PolAdv5)

> Have you heard of 'cadre deployment'? . . . Some of the senior provincial jobs have been members of parliament. (PolAdv3)

### Specialists' demographics

Considerations of 'race' also contributed to the diminished positions for PHM specialists. The use of 'race' here is as a social construct. The racist South African state, in the apartheid era, enforced and entrenched this construct in social institutions, such as education and health. It conferred privilege to 'white' people–education and employment–at the expense of SA's majority–'black' people who were largely seen as cheap labour.

As part of post-apartheid structural redress, policies to increase the appointment of black applicants through the policy of Employment Equity encouraged the appointment of suitably qualified black professionals to managerial posts in the public sector. As most specialists were 'white', the specialist requirement for these positions was removed to increase the pool to generic managers which included many more 'black' candidates. While this was recognised as a fair interim arrangement whilst a new cohort of 'black' PHM specialists and PH professionals emerged, it also had a downside:

> My move was lateral and downwards . . . I believe it included elements of an anti-white restructuring, as several other white managers were also demoted at the same time. (Other2)

With the transition, there was a sudden recognition that by specifying it as a . . . require-
ment, it would completely undermine black economic, let's call it 'black promotional'
empowerment. I think that one of the key reasons public health stumbled in the 90s [was]
because there were very few specialists of people of colour in public health. (RetrdAc2)

**The era of generic managers.** Some believed the anti-science stance of Dr Tshabalala-
Msimang amplified an anti-doctor sentiment that pervaded the health services in the early
2000s. This dynamic was also said to underlie the replacement of doctor-managers with
generic service managers:

And there's also this whole issue . . . the doctor-bashing syndrome. . . . "Just because you're
a doctor and you've got a four year specialisation, why should you be any better than me
who has been a nurse who has got an MPH?" (Priv1)

This resulted in the appointment of staff to run health institutions who did not necessarily
have health backgrounds. Informants commented on the poor calibre of managers recruited
and negative impacts on the health services. Whilst management training, through Master of
Public Health (MPH) and management programmes, were made available, many believed
these failed to upskill managers working in complex institutions:

There was a notion that anybody, any health scientist, even the non-health professional
could run a clinic, a hospital, be a hospital CEO, run the district health system, run a pro-
vincial health department or a national health department. (PolAdv5)

Managers in the health system needed more training and the Masters in Public Health was
seen as a route to that and that it shouldn't just be doctors. The level of the MPHs gave peo-
ple a better understanding of public health issues. But I don't know if it actually gave them
the competencies to lead and manage. . . changes that needed to happen in the health sys-
tem. (Acad4)

**Limited human resources.** Informants argued that PH skills-sets, underpinning health
system design, were scarce. The pool of PHM specialists was small and only a few new special-
ists graduated each year. Therefore, it was futile to earmark a category of jobs for PHM special-
ists as the likelihood of recruiting someone with a qualification was low:

People started looking at public health for . . . what are the main drivers and causes of ill
health and, the most ethical way . . . health system ration[ing], issues about health systems
management, issues about health systems design. As we became aware of the imperative of
those sorts of skills and health systems, we underst[oo]d how limited that sort of skill base
was within the country. (Priv2)

People carried on without them [PHM specialists], and not many applied and forced their
way into the public sector. . . This position that I'm in, is not for a public health specialist. . .
I have toyed with motivating to make it a specialist position, but then, you'd be blocking
recruitment. (ProvMan1)

In addition, examples were given of both pre- and post-apartheid government failing to uti-
lise available skilled PH personnel. One informant struggled to find employment because of
racism during apartheid, and another commented on the persistent inability of government to
draw on available scarce and valuable skills:

It took them a year to appoint me and then I was appointed as a medical superintendent with a [specialist degree plus] four years' experience in a rural setting. They appointed me with. . . a white female doctor, recently qualified. (Other1)

The tragedies of specialist medicine in South Africa, not just public health . . . is the failure of government managers to use experts. . . For example, you can have a collapsing obstetric service. . . The [name] Health Department is surrounded by public health faculties—and here you get a collapsing health department and people will not call on expertise. (NatMan2)

**The public sector work environment.**   Structural factors negatively impacted on public sector recruitment and retention of PHM specialists. A key strategy to retain medical specialists in general was the 2009 Occupational Specific Dispensation (OSD) which improved physicians' salary structures but it did not apply to those working as managers. Poor remuneration then became a disincentive to remain in management positions:

Because generally in the public sector, any senior management post that gets advertised, it's matric plus three years. People can get up to the level of chief director just with a three-year qualification and. . . all the health posts get written that way. (Priv1)

In addition, the public sector was not an easy work environment, with stressful work, competing demands and frustrations with an environment that constrained innovation. Interestingly, one participant questioned whether the hierarchical work environment in the state sector was suited to the innovative practitioners the speciality aimed to produce. These factors may limit recruitment and the public service as a long-term career option:

It attracted deeply motivated people and that made a huge difference. Many of those guys did not survive in the rigid uni-dimensional kind of structures–be they local; provincial or national. (RetrdAc2)

**Public Health Medicine and specialists' poor profile.**   Factors contributing to the speciality's poor profile are presented in Fig 2 and discussed below.

PHM is not focused on direct individual patient care, prized in medicine. Consequently, some believed that the speciality was not respected by physicians and argued that clinicians' opinions on PH issues carried more weight. This, together, with instances of mediocre work produced, contributed to the speciality's poor profile:

Public Health Medicine. . . was regarded as 'common sense', it was in fact stated by a. . . professor friend of mine. . . "I have no respect for your discipline". (PolAdv6)

With their gravitas as a Head of Medicine, they make comments about the national health insurance and they get listened to. The prof of Community Medicine doesn't have the same gravitas, . . . because is it[he/she] a real doctor. (Other1)

Clinicians looked down on it. "These guys, what do they know, they don't do the real medicine". Public health people . . . were doing pretty mediocre kind of work. (NatIns1)

Some felt that this historical 'poor cousin' status has shifted, with PH research gaining recognition, although ignorance about the speciality remained ubiquitous. Others thought PHM specialists self-identify in particular content areas, for example, they become known as an HIV epidemiologist or a Health Promotion activist or Maternal and Child Health expert, and do

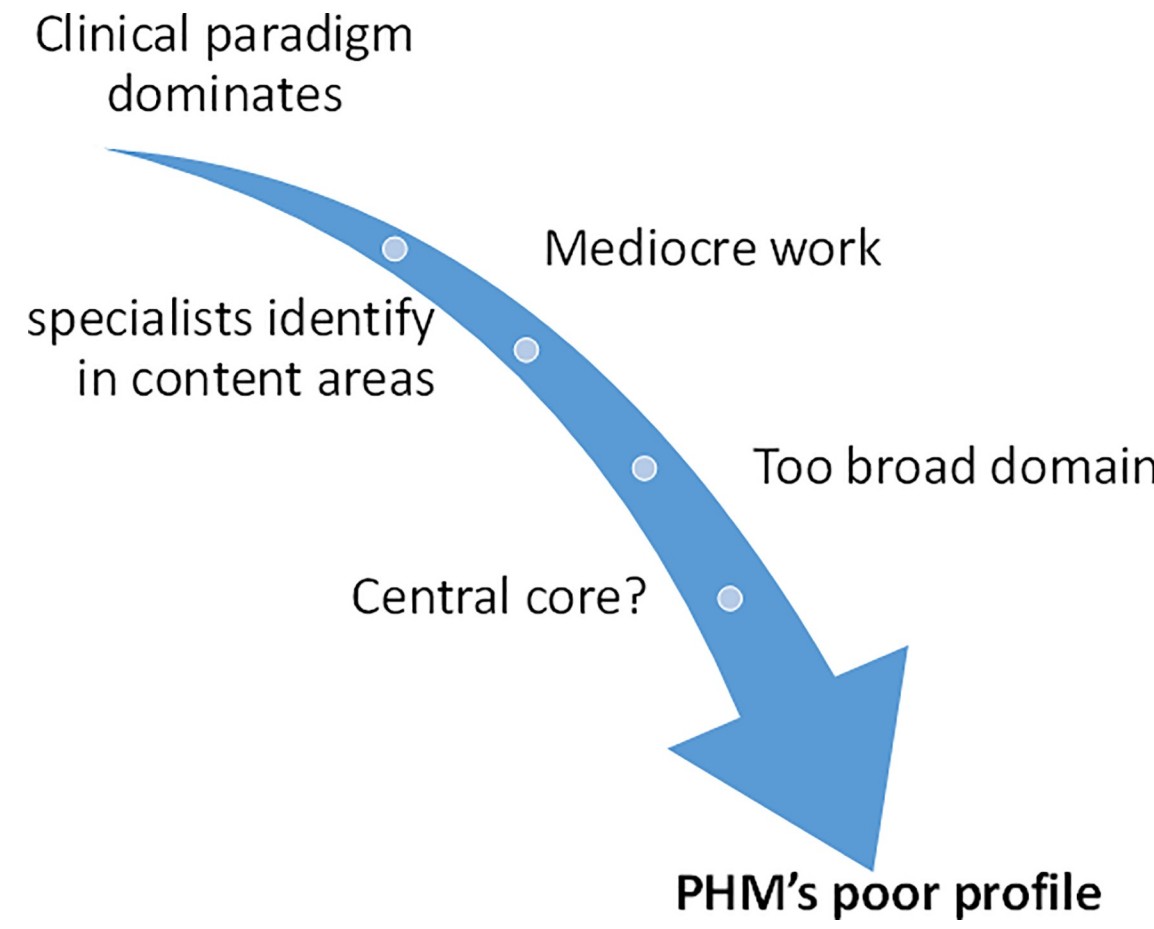

**Fig 2. Factors contributing to Public Health Medicine's poor profile.**

not publicise their PHM identity. Also, some believed that there was confusion about the difference between PHM specialists and Family Physicians as both are conflated as primary health care physicians:

> I didn't know that they [PHM specialists] existed. . . I don't think they, as a speciality, have sufficient prominence within . . . medical conference[s] . . . where you become aware of people. I don't know why because they do interesting work. . . Do they go to other conferences? (NGO1)

The identity of PHM, including whether something central underpinned the speciality, holding it together, was questioned by some informants. Linked to this was the sense that the competencies expected of the speciality are broad and, consequently, lacked coherence. Some argued that constituent disciplines such as infectious diseases or occupational health were easier to grasp because of narrower focuses:

> Perhaps part of the problem is that it really isn't a single discipline, you know–it's a conglomerate. . . It's a whole lot of things that can't be put anywhere else, that have all been put together for lack of knowing where to put them. (NatMan1)

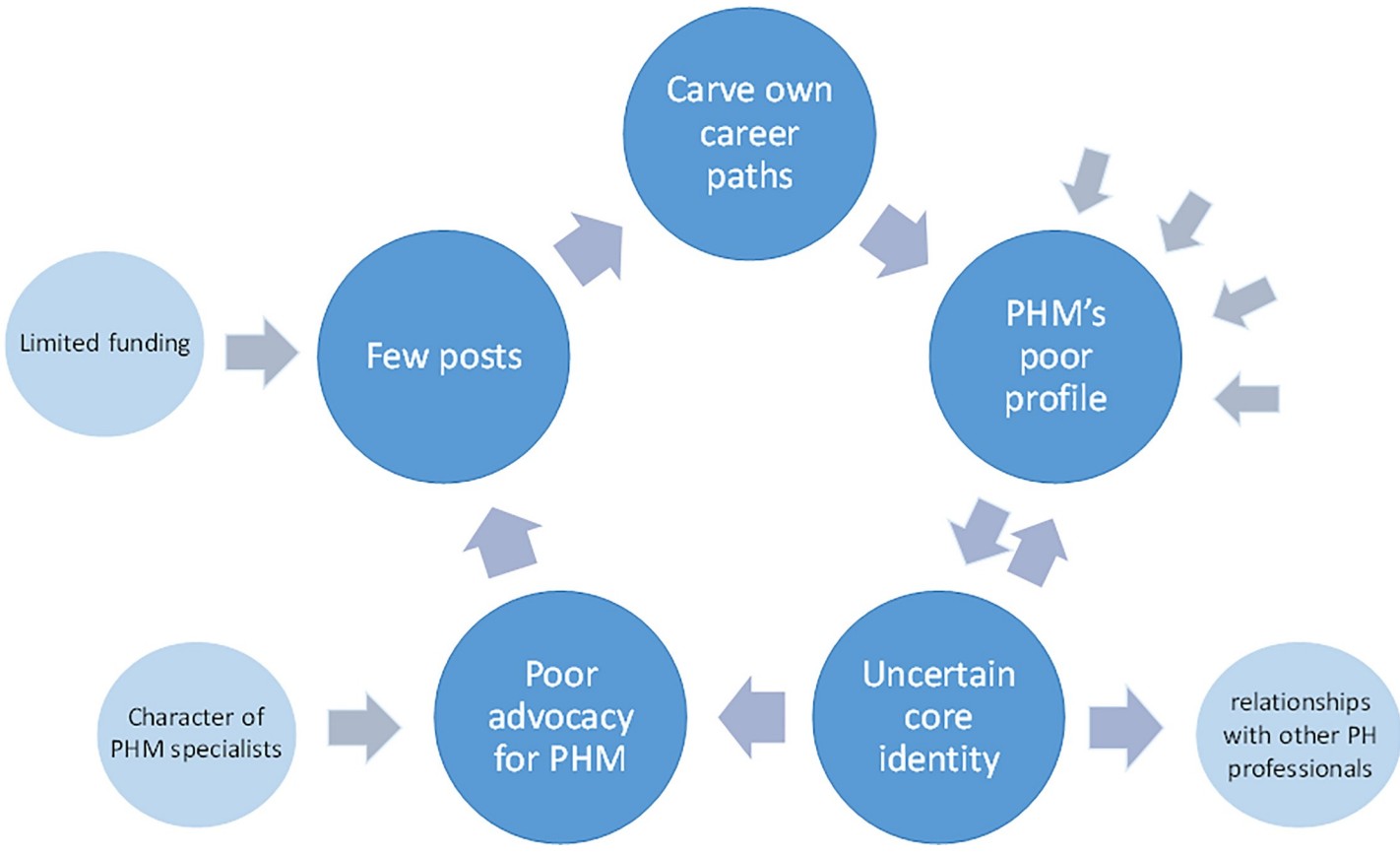

**Fig 3. Relationships between factors impacting on Public Health Medicine's poor profile and ill-defined identity.**

We haven't been able to articulate the umbrella. So we have recognition, medical and dental council, . . .fellowships, and that keeps us going. . . But take those away, we may have a problem. (Other1)

On the other hand, the breadth of PHM was also seen as advantageous, enabling the development of a versatile professional, who "know[s] a lot about everything", equipping physicians to work in a range of contexts at various levels.

The consequences of PHM's poor profile and contributing factors are depicted in Fig 3. An uncertain identity reinforces a poor profile, which translates into poor advocacy for positions, few posts and results in specialists carving their own careers. This in turn reinforce a poor profile. The four arrows represent the four additional factors impacting on PHM's poor profile given in Fig 2.

Informants argued that an ill-defined core identity resulted in poor advocacy for the speciality, which together with inadequate funding for training and specialist posts, were a major reason for PHM's poor prominence. Few posts in the public sector translated into unclear specialist career paths, and resulted in many specialists carving their own path:

There is not a clearly defined career path for public health specialist[s]. I'm going to blame public health specialists for this. I think those of us who have been involved in public health, we keep blaming the Department of Health; but what have we done? (NatIns2)

I qualified and then I couldn't find a common road that people take. That is why I said, "Alright, I'm interested in community projects where I will explore the role of health" . . . In other words, I carved my own path because there was no general path. (PolAdv4)

Poor advocacy for the profession and poor differentiation between the various cadres of PH professionals resulted in the speciality not being made a requirement for positions. PHM was not marketed as central to the work of the services while Family Medicine was proactive, securing a service niche for itself. However, contextual factors such as the development of the district health system, which prioritized clinical primary service delivery and required clinicians, contributed to Family Medicine's success:

They haven't differentiated a specialist doctor in public health, so that's part of the reason why it's never been put down [as a requirement for a job]. And it's also more difficult for the people like me to advocate for it because then it's seen to be personal. (Priv1)

It is difficult for public health to do a marketing job because public health's presence is within occupational health, environmental health, epidemiology, research projects; and not actually assisting and implementing primary health care. So it's come back to hit us. (RetrdAc4)

The character of the PHM specialist may also be implicated in the lack of advocacy for the profession in the services. They were profiled as focusing on people and not on the profession. Self-effacing specialists were reluctant to alienate others and consequently had not promoted the profession. Indeed, some found past lobbying for recognition for the speciality–for specialist posts and overtime pay–to have backfired and was distasteful:

We are not arguing types . . . We tend to be nice and gentle and loving and kind and must think of the whole of humanity. (Other1)

It was distasteful, you know. Certainly we need to make sure that we are remunerated appropriately and we have jobs. It was not coming from a perspective of public health specialists can add value to the health system and they have a unique contribution to make. So, I think, that kind of lobbying around overtime . . . almost prejudiced Public Health [Medicine] specialists. (Acad4)

**Uneven training programmes.** The quality of PHM training was contested. Some believed that that the nationally accredited exit examination required successful graduates to operate as intelligent, autonomous individuals. This, combined with broad skills-sets and apprenticeship-type training, fast-tracked them to work in a range of organisations better than other PH professionals. Informants commented on the high calibre of specialists produced and believed training was pivotal to their leadership positions in service and academic institutions:

Give me a person who's got bio-medical sciences as his or her base and you add to that the social sciences, sociology, political sciences, economics. . . the management stuff . . . environmental sciences and the occupational health stuff. . . all in one person. . . that's a unique [person] that can probably thrive anywhere. . .You find the health sciences faculties' deans are Public Health Medicine people . . . at some point they rise to the top. . . I think there's something public health gives you. . . (PolAdv6)

However, a few believed that some physicians had selected the PHM speciality as work was confined to office hours and was not demanding. In addition, a view also expressed was that training was sub-optimal in some institutions, with registrars (trainees) focussing on academic work, adding little value to services. One dissonant view signalled a concern that some training programmes produced practitioners that were not fit-for-purpose:

> People did do it for an easy ride, you know. It's easy to do public health and the hours are controlled. (RetrdAc3)

> They seem to be full-time students rather than like other registrars, basically because they're working from the university, getting paid on a full-time basis and their value isn't realised. (Priv1)

A few had employed specialists who were unable to lead and manage. This was partly due to personal attributes of registrars, but also was attributed to gaps in some training programmes and the PHM curriculum–particularly in relation to project management, monitoring and evaluation, health economics, and leadership:

> A lot of stuff around monitoring and evaluation, impact analysis–and our training hasn't been as good in that. But if you want to work at the national Department of Health, that oversight role is one of the most important. (Priv1)

> Hard modelling skill was not something that came through in public health for me. But when I came into the service, it was a big thing. How do you model, projecting, forecasting. . ..? Project management is a big skill. I can't recall doing project management in my public health training. (ProvMan2)

These critiques about PHM training highlight the importance of strong theoretical programmes coupled with practical training involving placement of registrars in appropriate service settings. This, together with academic supervision and assessment, should enable the development of strong and relevant competencies.

**Relationships with other public health professionals.** Together, an uncertain identity, poor advocacy and the breadth of the discipline resulted in blurred boundaries between PHM specialists and other PH professionals, such as MPH graduates. Opposing perspectives about the nature of the PH workforce underlay reflections about the niches of various PH professionals. The few with hierarchical perspectives, envisaged PHM specialists occupying top echelons, with MPH graduates being subordinate. In a context of competition for jobs, MPH graduates could be a threat to specialists. In contrast, those envisaging a flatter system with complementing contributions from PH trained staff, believed that arguments about differences between various PH cadres alienated decision-makers and was detrimental to the profession:

> What is happening is people doing their MPHs are claiming this space that we have. That is a problem. (Acad5)

> The MPHs and the MPhils, that has helped to produce another cadre of people who understood epidemiological theory, and they are applying it. (Other1)

For others including non-PHM specialists, MPH graduates were a heterogeneous group, whose interests and skills depended on their base profession and training and they did not threaten the PHM space. They used the MPH degree for career progression:

Most of them [MPH graduates] have advanced themselves in their own particular fields. So environmental health officers . . . have just progressively risen in their fields. (Acad6)

The range of MPH degrees available, with variable foci and lack of accreditation, made their value uneven and contested. While some thought MPH graduates added value to the health services, others doubted the training adequately prepared graduates for work. Although MPH training aimed to produce graduates with generic management or focused technical skills, teaching standards were questioned. Others believed competencies expected were unrealistic and MPHs cannot produce managers to lead and manage change. These skills were better achieved through practical exposure:

There is quite a strong focus on the implementation cycle, so that when graduates emerge, they're supposed to have the skills to do situation analyses using quantitative and qualitative methods, to plan comprehensive programs and to be able to monitor and evaluate it and also to know something about how to advocate for and communicate about this. (RetrdAc4)

People have a MPH behind their name but they cannot apply any public health, and that was one of the presidential directives to capacitate health managers, "Do MPHs!" So, this is an instruction, to take on MPH students. . . They were just passed, without decent dissertations, with useless marks. (ProvMan1)

I think it [MPH] adds some value, but it's not [for] people who are going to run the health services. I think you have to have something more substantial. . . the epi[demiology] skills are good, but their managerial skills are zip [nil]. (PolAdv3)

## What is needed in the services?

Having explored responses about PHM's absence in the services, we probed the needs of the health services in an era of reform. There was agreement that there was a need for competent personnel to translate articulated goals into service delivery.

All identified the need for leadership, management and planning skills, and some added PH perspectives and skills. They believed that the scarcity of competent managers with health backgrounds, heading complex institutions, and with project management and planning skills, compromised the services. Gaps in human resources management, policy implementation, research and report writing skills fostered dependence on consultants with strategic skills such as change management.

There is a general inability . . . to translate vision. . . into a strategy that can [be] implement [ed]:. . . broken down into an operational plan that says clearly who's going to do what, by what date, linked to a budget that supports that operational plan and that's underpinned with an appropriate staffing structure and an appropriate and strategic information framework that allows you to track progress. That's what's missing. (NGO1)

It's absolutely shocking how poor people are at report writing. And to be able to . . . document and research and come up with new ideas. . . . Then you get these consultants that come in and take all your stuff and then put it into a fancy report. . . and then suddenly they're the experts. (Priv1)

Informants believed that managers with PH skills could lead, plan and manage complex services and institutions as PH is grounded in understanding the health profile and needs of populations, and facilitates critical engagement with policy and practice:

People with strong change management competencies, to address not just routine delivery of services, but to have a broader vision of prioritising, burden of disease, quality of care–a higher level of expertise in terms of planning, evaluating and developing the services. So there has been a huge gap in that kind of leadership and we've seen . . . the results of that in terms of the quality of care that's being provided, [for example] the [disease] outbreaks . . . which shouldn't happen. (Acad4)

There are thousands of people out there who need some substantive public health skills. . . They need skills to be able to run a district or run a program and they don't have that. (RetrdAc4)

Some applauded the present health ministry's efforts to remediate the past 'generic management policy' and believed that this move held hope for improving health services:

Dr Aaron Motsoaledi, the current Minister. . . publically bemoans that period where the Department of Health went through a phase of appointing non-professionals to very senior [positions] . . . and he now wants to turn that around. (PolAdv5)

The PH unit in one provincial office was valued. This structure, housing PHM specialists, promoted PH perspectives and built capacity to manage the system as a whole:

Our whole big push in this province now is to just get everybody thinking about the system as opposed to your own patch and that's a whole mind set change. (ProvMan2)

## Roles for specialists and public health professionals

The importance of PH trained managers in health institutions raised questions about the need for PHM specialists, and management competencies in PHM training. We probed perceived differences between competencies and roles of PHM specialists versus other PH professionals with MPHs or Masters in Business Administration (MBAs) so as to surface a particular PHM identity.

Informants believed that PH skills, such as epidemiology–which underpins data interpretation–had to underlie management roles and the robust policy and decision-making required for health reform:

But you can't get involved in management sciences if you don't have a foundation in epidemiology. (PolAdv3)

In an era of health systems reform, many believed there was a place for PHM specialists who had deep PH competencies and management skills, both key to service transformation:

Public health specialists need to be [focussed] on . . . how do you reinvent service delivery? How do you deal with the failures in service delivery? Unless I'm mistaken that is what public health is supposed to be doing. Because there is no-one who can do that. (NatIns2)

Whilst they believed PHM specialists could make a contribution, some questioned the competencies of some current specialists who did not have the required management skills. Others argued that the Health Ministry did not recognise the broader value of PHM specialists who had expertise for health reform. It did not harness the wide experience of those already employed in management positions:

The Public Health Medicine specialist could really be an incredibly powerful force for good. If they came out of that speciality [training], understanding the public health issues, but also understanding the managerial competency that goes with it. I'm not convinced that that exists. (NGO1)

When I came here as the public health specialist with a glittering CV, a lot of experience, they still closed the door on my face. That made me think that . . .[the] public health trained doctor is not relevant. (PolAdv4)

They saw that staff with MPHs had 'technical' skills, for example, working with databases, surveillance, in health prevention and promotion programmes. Some believed that pro-grammes such as child and women's health required MPH graduates with health backgrounds and argued that middle managers with technical responsibilities required a PH qualification:

If you're looking at child health, women's health, . . . health services, you need to have an understanding of health in general. I would think that everybody above the D[eputy] D [irector] should have an MPH of some sort, or equivalent. (RetrdAc1)

Most questioned the advisability of promoting MBA qualifications. They believed these programmes imparted skills that were inappropriate for state health services, which required tailored skills such as project planning. In addition, they believed physicians with MBAs were rare and would move to the private sector:

I am very reluctant to punt MBAs because I think you can put a hell of a barrier in there. I think it's easier to get people through. . . postgraduate certificates or diplomas and teach them the basics about how to do a budget, how to do a marketing plan, how to do a human resource plan. MBAs cost a lot of money,. . . a big price to charge for professional health management. (NGO1)

[The] medical administrators group . . . they were running those hospitals. . . .When the budget crunch came, they had MBAs and the private hospitals just snapped them up. (Other1)

### Developing the Public Health Medicine speciality and identity

In view of the ill-defined difference between PHM specialists and other PH trained personnel, informants argued that the speciality's leadership should re-examine its competencies and identity to craft a way forward in relation to service roles and clarify its relationship with the medical profession.

I think there is quite a need for introspection by leaders of the profession and the kind of rethink of where we want to position this profession going forward. Does it still have role? What is this role? What do we actually want for Public Health in South Africa? (NatMan2)

The issues that emerged under this theme are depicted in Fig 4.
Although many believed that it was timeous to rethink the profession's identity and future, some felt uncomfortable about discussing and promoting PHM apart from PH more generally. Many maintained PHM specialists should be part of teams working with other PH trained pro-fessionals, often in leadership positions. Interestingly, one argued against defining a unique competency for PHM specialists as the strength of the speciality is its breadth, and this would contract its area of work:

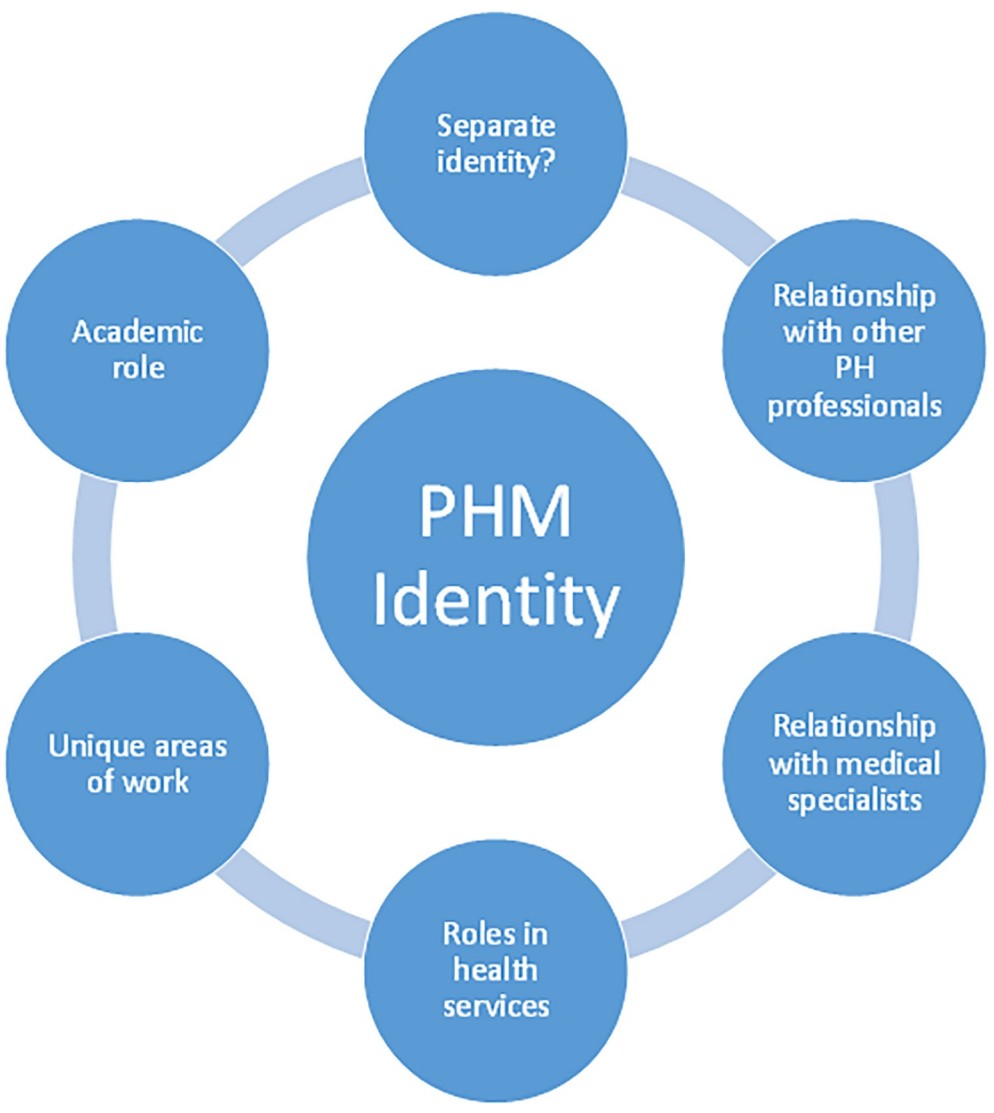

**Fig 4. Issues related to Public Health Medicine's identity.**

I'm not somebody who feels comfortable with the sort of self-advancement–pushing the profession of Public Health Medicine above all else. . . In the ideal world there would be a team. . . I think that the team needs to have both MPH trained and MMed trained [Public Health Medicine] public health [professionals]. (Acad3)

If you say that these are the things that only a medical doctor in the field is competent for, you start contracting the field of practice and skills down to this minute area of narrow competence which I think undermines the discipline and the expertise. (Acad1)

**Public health roles in the services.** Many were encouraged by the mention of PHM specialists at all levels of the services in national policy documents. We probed functions for PHM specialists and PH trained professionals. Roles suggested spanned all levels of the health services, from national to district, in leadership and support functions and new, statutory national

agencies such as the Office of Health Standards Compliance (OHSC) [16] were opportunities for PH graduate employment. Some, however, also believed that senior positions required a PHM qualification:

> There is a glimmer of hope. What we think is our main function, and it surfaced in the Human Resources for Health document, is in the establishment of public health units. (Acad5)

> The National Health Amendment Bill (OHSC) it is now an Act of Parliament. It's non-negotiable. Nobody can deliver on core standards except public health people. . . I mean MPHs and specialists. So, there's enough work for us for the next fifty years. (PolAdv5)

There were comments about roles at a national level, and a few noted the lack of PH trained staff in SA's national Department of Health. Some felt the priority was provinces with roles in monitoring, evaluation to inform service planning. It was important to consolidate scattered expertise into provincial units, with capacity in epidemiology, in evidence-based health informing decision-making, employing personnel including PHM specialists:

> At provincial level, because I think that is where you need to elevate the role of public health specialists. If you can get it done there then I think strengthening the districts is going to be a lot easier. (NatIns2)

A similar wide range of functions was suggested for PHM specialists at district levels, with roles in surveillance and planning, assisting prioritisation, operational research, resource allocation, service design and implementation. For some, MPH graduates could do this technical work. Many advocated that PHM specialists should be part of district management teams or be district managers. One proposed the introduction of regulations specifying PH qualifications for district managers, and requiring existing managers to obtain one within a specified time:

> They could play a role in district planning, implementation support, operational research, M&E [monitoring and evaluation], Epi [epidemiology], disease outbreaks. What we need is public health capacity to analyse the data, to provide the intelligence, to inform our decision making. . . How do you set priorities, what is the research data? Implementation support, operational research, you know, rapid appraisals to see whether things are working, not working, all of that. (ProvMan2)

> So it would be great if the district health manager had public health skills and maybe we would want to get it in some kind of legislation that every district health manager should have done. . . an accredited course in public health. (Acad2)

**Public Health Medicine and clinical medicine.** Many raised the importance of PH perspectives for all physicians, including medical specialists. More prominent PH content and perspectives in medical training could translate into an appreciation of the value of PH, PHM and specialists' work, and entice students to specialise. One suggested that PH competencies should be core to other medical specialties, as PH perspectives add value to individual patient management. Medical training lacked PH approaches, which consequently requires reinforcement in specialist training:

> You don't ask them under what conditions do they live. . . .So I think [public health] should be part of everybody's speciality. Whether you're a cardiologist, or whether you're a

paediatrician, you should have a public health slant to your work. And that probably is the single biggest weakness in specialist training. I'm not knocking that we need specialised public health doctors. (NGO1)

For some, the value of medical training for PHM specialists was understanding bio-medicine and their prior clinical experience in the services–all useful for population orientated work. For others, being a 'specialist' could enable working in teams with other medical specialists managing services:

It does help to be a medical specialist if you [are] engaging with public sector managers. . . there's still a certain level of respect that's given to the opinions. . . that probably makes them a bit more effective than the person with just a Masters in Public Health who might have been a physiotherapist. (NGO1)

A few argued that bio-medical and clinical training could point to specific work for PHM specialists–in clinical epidemiology and infectious diseases which could include clinical work. Statutory requirements for communicable diseases control, for example, was raised by another, which could be a work domain for PHM specialists:

An area in public health where you need to have an understanding of the human body in order to function optimally is Clinical Epidemiology where you are clinically competent and you are epidemiologically strong; . . . [and] an infectious diseases specialist that can manage patients in the ward, outbreaks . . .You're completely competent to deal with the biggest burden of disease. (PolAdv6)

We haven't got. . . the statutory functions to oversee surveillance, for example, and to assist with notifiable diseases. The same with port health,. . .haemorrhagic fevers, the SARS, H5N1, all these things,. . . could be part of our domain. (Acad5)

## Academic Public Health Medicine

Many thought academics were the 'face' of the profession, driving its direction. As academics were removed from the health system, PHM was largely absent in health policy and implementation. In addition, some believed the small number working in academia and quality were insufficient to give direction to the speciality:

A weakness in the profession is that many of the public health professionals are more comfortable on the university turf and don't actually want to come into the services. (NatMan2)

There are not people who have the academic criteria and who have the leadership potential and the vision and the people and vision skills to actually do that. (Acad6)

Poor relationships between some universities and provinces, had resulted in impasses and service agreements between them were not signed. This jeopardised academic 'joint' specialist positions and the future of academic PHM. One respondent noted that, as a result of this disconnect, 'joint appointment' academic specialists were not replaced:

There is no [joint management] agreement. . .There is nobody there to motivate that they [posts] be filled . . . I'm really worried that there won't be a Public Health Medicine department in five years. We may [be] subsumed by Public Health. . . Public Health Medicine, subsidised and created MPH and now that will actually take over. (Acad6)

'Joint appointment' academics voiced difficulties with juggling responsibilities for teaching, research and service. For some, heavy teaching loads meant service work and research took backstage, and consequently career advancement was a challenge as they needed advanced degrees:

> The reality is that you won't be appointed as head of department whether you were absolutely the right person, because you don't have a PhD. (Acad6)

> You just get lugged with all the teaching and undergrad teaching and supervision and you've only got that limited amount of time. [University employed] academics have much more free time to do grant writing. . . and to publish than we [joint appointment academics] are. Yet we are judged by the same standards. (Acad5)

Historically, service responsibilities for academic PHM specialists were to conduct special projects that were not necessarily embedded in the health services. This was unsatisfactory, and caused tensions, particularly as provinces funded these academics:

> I think public health from a service perspective was . . . neglected, and I think that's why there's this big schism between public health and public service. (RetrdAc1)

However, a few articulated a fresh approach with academics becoming more inserted in services, assisting with health priorities and strategies to improve service delivery. One provincial manager believed registrars' work and the current presence of 'joint appointment' specialists in the provincial office made the value of the profession visible:

> The consultants are now more and more getting bedded down in the service and contributing, feeling the issues, forging the relationships and bringing their expertise to bear and that's become more formalised. (ProvMan2)

Many described PH research that could inform health service delivery, including implementable clinical interventions, health systems research and building research capacity among service staff. In addition, for some, academic affiliations enabled sufficient distance for research and more critical contributions. Relevant research could give both credibility to the profession, strengthen the health services and result in the services viewing academic institutions as a resource:

> My research had swung completely to doing epidemiological work. I felt a certain frustration; I kept describing things. . . So I went to the university and brought around me. . . clinical people, pharmacy people. . . We're not really in the business of counting anymore; our focuses are on solutions. (NatIns1) helping people in the public health service to formulate the questions that can answered by research and academia. (PolAdv2)

> I think academic institutions . . . need to be seen as independent objective critiques. . . to try and move the health system. (Acad4)

**Teaching public health.**   PHM academics teach medical students (who could later enter PHM training) as well as others enrolled in post-graduate PH training. In this context, informants' perspectives on PH and PHM education were probed.

Many believed all health science graduates should have skills to engage health systems and policies. A range of postgraduate courses should be tailored–from introductory to advanced–to upskill health services staff:

> A challenge for curricula in public health, both at an under- and postgraduate level, [is] to teach more about health systems, about health policies. (PolAdv2)

> [Postgraduate teaching] needs to be differentiated. [Institutional name] has . . . public health [courses] aimed at. . . people who work at policy level. We want to develop public health practitioners for the operational level. . . You can have hundreds of different course units which would fit. . . people at different levels. (RetrdAc4)

As specialists' career paths were also in research, the private sector, occupational health, as well as self-initiated work, a few believed PHM teaching should prepare physicians for these options:

> There isn't a single career path but you have lots of career paths. You can go to the MRC, become a competent researcher. You can go work in the health services as [a] health manager at any level. You can go into the private sector . . . You will have skills for. . . those areas with your basic training in Public Health Medicine. (PolAdv6)

### Locating the speciality's future within a changing health system

Informants recommended improving the profile of PHM, and its service impact. Processes outlined to assure this are depicted in Fig 5 and detailed below.

Informants believed that health service reform was an opportunity to engage government and embed PH and PHM in the health system. National discussions about skills-sets of hospital managers and the NHI were opportunities to raise policy and service design issues:

> The Minister talking about people need[ing] management qualifications. . . and leadership institutes. . . . It should be a very favourable time for you guys to push your speciality. (NGO1)

They identified roles for PHM within the NHI–in policy development, monitoring and evaluation, and commissioning services. Furthermore, PH professionals' involvement in NHI service design could ensure preventive and promotive focuses are given due importance. A few remarked that the speciality should identify the skills required to work at a senior level, and train specialists accordingly:

> About NHI commissioning: You can't do that without understanding how you measure outcomes. . . the need will be there in five years' time, but who is positioning the speciality? (NatMan1)

> In many countries, national health insurance often actually ends up remunerating the clinical . . . and is not very good at public health. (NatMan2)

**Network and advocate for Public Health Medicine and public health.** In view of the dearth of PH staff in departments of health, many recommended that a network of PH professionals and specialists could contribute to a public sector focused on population health outcomes:

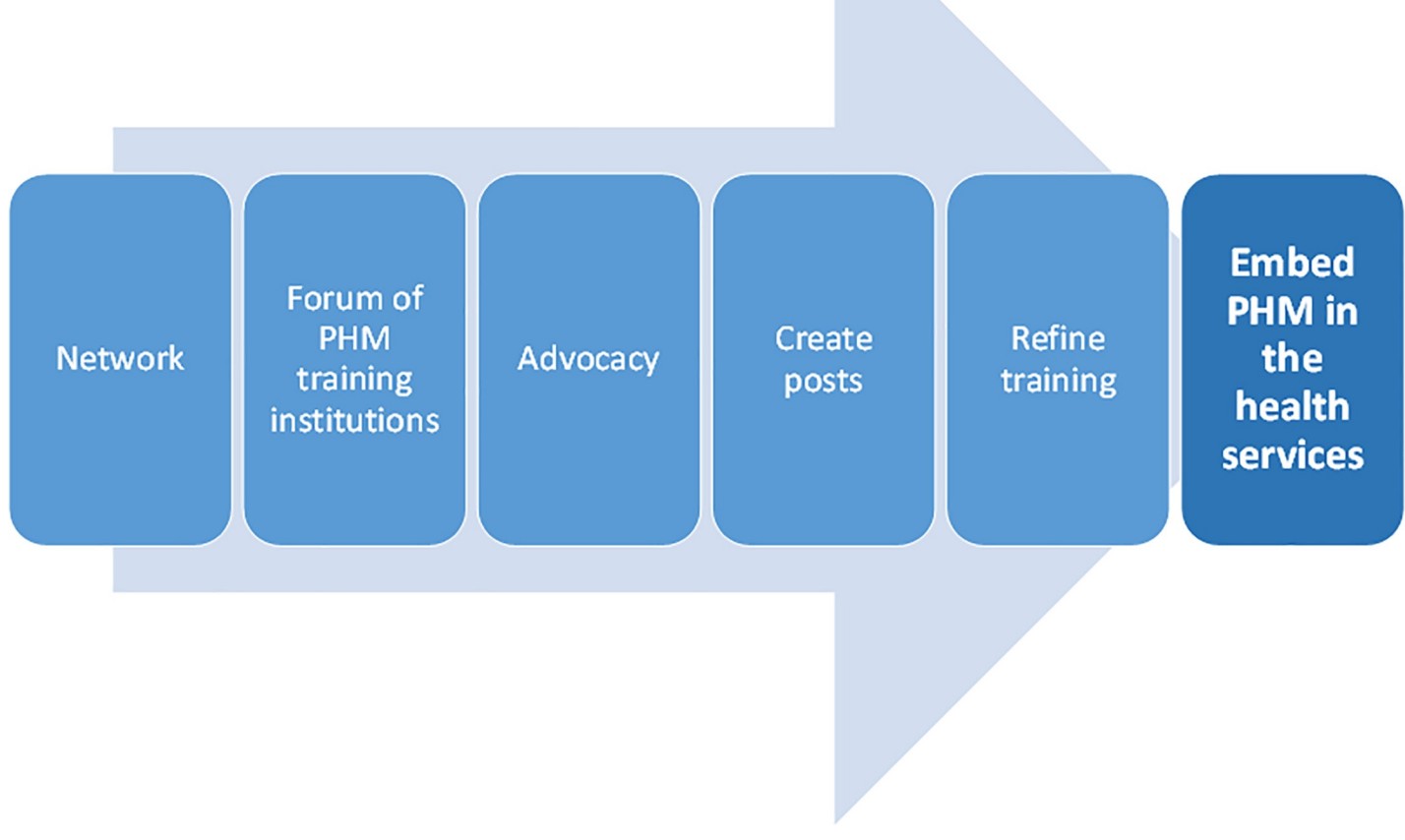

**Fig 5. Processes involved to embed Public Health Medicine in a changing health system.**

> It would be quite powerful to set up some kind of a group of public health people working in Government. . . a network. . . to defin[e] what you want to achieve and what are the strategies. . . how do you actually impact on population health. (NatMan1)

Within that, many recommended a forum of PHM training institutions to develop a common vision for the profession, refining roles and identifying PHM's place in the services and engage government. Discussions should develop trust and facilitate working relationships. Some warned against merely asserting possible contributions and listing competencies. Rather, PHM specialists' value should be demonstrated through showcasing functional PH units at provincial and district levels–their engagement in policy, planning and implementation issues. The forum could engage national and provincial health ministers, the National Health Council, together with professional organisations such as the Public Health Association of South Africa (PHASA) and the College of Public Health Medicine (CPHM):

> I think there needs to be a more active forum that engages. . . and maybe there is a discussion between public health and the Minister, to say: "We understand all of these priorities. How can we support you on NHI?" You see, the Minister is very open to this idea of doctors. (NatIns2)

We need to collectively document [public health units] and make a case study and suggest to all the other provinces that that's the way to go. (PolAdv5)

**Create posts.**  This process could lead to the creation of service posts for PHM specialists and PH professionals within PH units at provincial and district levels:

Ring-fence earmarked specialists' space, posts that are a career path for . . . Public Health Medicine specialists. . . And then we should have posts which are open to anyone . . . in the broader public health discipline. (Acad1)

Whilst most believed the cost of creating PHM specialist posts would be prohibitive, one policy-maker thought this would not be difficult provided a convincing business case was made to the Ministry of Health. Others cautioned against high expectations of only a few specialists in posts who could not perform a full range of PH functions. Additionally, some maintained that specialists working in the services were role models for new specialists and attracted people to specialise:

If we wanted to create two hundred public health posts in the public sector we could do it. . . If the Minister of Health said to us in the next three budgets "Look we really want to build the public health discipline". (NatMan2)

The full spectrum of Public Health Medicine's skills and competency can't be delivered by just one or two Public Health Medicine specialist posts. [if] we ask for that we are going to set up ourselves for failure. (PolAdv5)

Many believed that not all specialists working in the services should be paid on high, specialist pay-scales. It would be appropriate and valuable for newly qualified specialists to work as managers in districts and provinces:

Be prepared to work in management and then the money question comes in. . . They would need to spend some time in district management or provincial management in order to get that understanding. . . of how the state functions. (NatMan1)

**Refine training.**  Informants made important recommendations for revisiting the training for PHM specialists, MPH and health sciences' students.

They argued that new PHM service posts must be filled by qualified and able specialists, which had knock-on implications for trainee recruitment. Registrar posts should increase to cater for an anticipated demand. A few believed that applicants should demonstrate prior social and service commitment, and leadership abilities. One proposed that senior service physicians should be head-hunted for training:

You've got to have a social activist background to do it. You've got to have good epidemiology, so you're not just being an activist. (RetrdAc3)

Head-hunt people. You've got to go and get a guy out of the services and say "We've been spotting you [], you have been ranked the top CEO in the country". (PolAdv3)

Some argued that the PHM curriculum should be reviewed, addressing skills gaps, to ensure that graduating specialists have appropriate high-level skills. Competencies proposed

corresponded with what were seen as service shortages–leadership, management, health economics and research:

> We can graduate specialists, with a pretty moderate level of knowledge and capability.. . . We've got to set higher standards. We want people who can optimise excellence in our field. (NatIns1)

> People don't plan their resources. . . they leave it to the economists to do it. . . We might be teaching it, but I think that's a vacuum. The strength is epidemiology; the weakness is management and planning. (PolAdv3)

As MPH curricula and graduates were hugely variable, some proposed accrediting curricula through an external body outside of the training institutions, such as a professional PH organisation:

> [PHASA should have a] permanent structure and eventually one would also see that it would become the accreditation structure for courses in public health. The structure shouldn't fall under the schools because. . . they can't be player and referee at the same time. (Acad2)

Informants believed PH should be prominent in health sciences students' undergraduate training and integrated in clinical training within service settings, with small research projects. These would inspire students about PH, impart skills for future practice and strengthen services:

> Undergraduates, they know clinical medicine. . . One would use that as an advantage . . . We would place them at facilities and let them do smaller research projects. . . Inspire them to want to improve and strengthen the health system when they qualify. (PolAdv4)

> We could and should overcome this separation between ourselves and clinical medicine by embarking on joint teaching, joint projects, instead of being sequestered off. (RetrdAc4)

## Discussion

In a context of health system reform in SA and unclear career paths for PH graduates, we explored the perceptions of the value and place of PH and PHM among experienced, senior informants in SA's health sector. Despite deficits in the past and present, all valued PHM and its competencies. Key skills highlighted were the interpretation of information from a range of sources and utilising these for decision-making, which are congruent with PHM and PH skills also noted in SA health policy documents.[7, 17, 18]

### Public Health Medicine's absence

We re-categorised the factors that account for PHM being largely absent in the services in Figs 1 and 2 into external contextual factors and issues internal to the speciality, as is depicted in Fig 6. This separation identifies issues requiring attention that are independent of and dependent on external factors.

PHM's absence in the health services was a combination of external factors, such as legacy of an inordinate focus on clinical personal health services, a global phenomenon,[19] in addition to political factors (politicians' promoting their own approaches; anti-doctor sentiments;

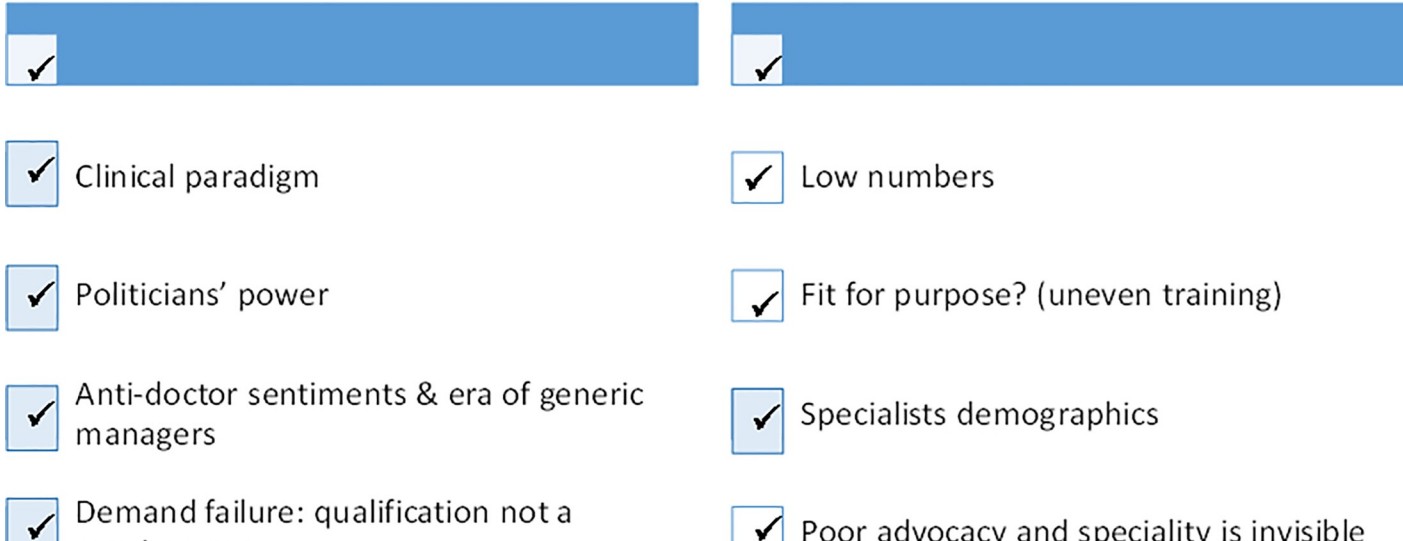

Fig 6. **Internal and external factors contributing to PHM's low profile in the health services.**

and era of generic managers), as well as specialists past demographics that required redress. Additionally, the speciality's profile was the result of 'demand failure' due to health legislation that no longer required public health qualifications for service positions.

As in SA in the 1990s, the introduction of generic managers formed part of health sector reform in New Zealand,[20] Australia,[21] and the UK.[22] 'New public management' was based on the notion that a professional ethos was at odds with sound management practice and inconsistent with efficiency and effectiveness. As was highlighted in reviews of this approach in the UK, [23] and by research among SA hospital CEOs,[24] our informants remarked on its negative consequences for both service quality and PHM specialist careers. Commentators on SA's health system also noted that this approach resulted in weak service management.[25] Informants beliefs that inadequate remuneration, bureaucracies and stifling of innovation negatively impacted on retention of PHM specialists in the public sector, is mirrored in studies that demonstrate autonomy at work and supportive work environments ensure retention of professionals.[26]

Internal factors included PHM's invisibility, in that it was largely unknown amongst professional and lay audiences, and was historically poorly regarded by medical colleagues. This is an enduring theme, dating back 25 years when PHM was characterised as a stepchild, the 'Cinderella' of medicine in SA,[27]. Similarly, a study of PHM specialists' identities in New Zealand highlighted the perceived poor value of the specialty.[28] This is changing in South Africa,

with PH becoming core to undergraduate medical training in SA,[29] and academic life, which is an opportunity to inform colleagues and future physicians about the speciality. Two leading universities–the University of Cape Town (UCT) and Witwatersrand (Wits)–both with long standing PH and PHM programmes, foreground PH in their research agenda and outputs, which speaks to the robustness and impact of academic PH and PHM. In 2015, UCT's Faculty of Health Sciences long-term (to 2030) vision commits to addressing upstream determinants of health, and strengthening health systems as part of its strategic theme of community and population health.[30] At Wits, the Dean of Health Sciences highlighted that the School of Public Health contributed the second highest number of publications in the faculty in the 2016–7 research report.[31]

Specialists' wide employment and roles contributed to PHM's invisibility and uncertain identity, its broad and sometimes superficial skills-sets. Globally, research on PH personnel and specialists also show unclear roles and insertion in national health systems.[32] An uncertain identity was also found in research from Singapore which found that PHM was "difficult to characterise" due to its intrinsically context-specific focus–improving population health, and multi-disciplinary orientation, with competencies defining professional status and roles. [33] The recognition of the contextual nature and broad scope of PH practice led the UK speciality to incorporate non-medically trained professionals,[34] and it is surprising that a multi-disciplinary speciality has not been adopted elsewhere.[35] Interestingly, no informants raised the desirability of professionals outside of medicine joining specialist training in SA. This may be the consequence of the examining body–the CMSA–being reserved for physician membership. However, in SA, PHM's broad scope must be recognised when defining its role, identity and relationship to other PH professionals.

Poor advocacy for the discipline was an important internal factor that contributed to ill-defined career paths and PHM's poor prominence in the services. Advocacy does not mean asserting PHM's exclusive contribution, but demonstrating value though registrars' and specialists' work while acknowledging other PH professionals' contribution. Current health system reform initiatives–enhancing the skills-base of district management, training appropriate health managers, the OHSC,[36] NAPHISA and the NHI–are opportunities to advocate for, reshape and insert PHM into the services.

**Training and competencies.** Many academics trained both MPH and PHM students and, while MPH programmes boosted graduates' careers, informants were concerned about the depth and unevenness in MPH training, particularly in management sciences, which was also noted by commentators about American programmes.[37] In contrast, PHM's unique training fast-tracked people to work in complex environments at leadership level, roles of PHM equivalents in the UK[38] and Canada.[39]

Critical requirements for health service reform highlighted were in leadership and management, particularly policy and operational management, which are also reflected in commentaries about SA's health system.[25] As identified in local literature as valuable for provincial and district level management,[40] PH skills are in epidemiology, combined with management, with MPH professionals adding technical skill.

**Repositioning Public Health Medicine in the context of invigorating public health.**
Consistent with international literature,[41] informants believed it was important to differentiate PH specialists from other PH professionals to ensure PHM's future. The PHM specialist cadre differed from other PH trained professionals as they had a broader set of competencies based on prior bio-medical training which, coupled with registrar training, facilitating a broader and deeper knowledge base which enhances the experiential learning, leading to a versatile professional with 13 years of study and professional experience. Other PH professionals

usually have undergraduate training in health or social sciences, and two years' theoretical PH masters' degrees, often completed part-time.

There were divided opinions as to whether other professionals threatened the PHM specialist niche. Those with hierarchical perspectives believed that PHM specialists should lead PH teams and MPH graduates made the speciality precarious by competing for the few existing jobs. Hierarchy in PH practice is a feature of health systems elsewhere, for example the UK enumerates nine levels in the health services.[42] Those appreciating the PH's multidisciplinary nature did not believe that clear professional boundaries should be delineated, as this contradicts PH's multi-disciplinary nature, multi-professional practice,[41] an identity of PH internationally.[43] Valuing the multidisciplinary nature of PH should mean that the SA speciality should debate opening up training to other professionals, unless roles requiring clinical insight and competencies become core to the speciality.

In our study, the value-add of medical training was the resultant biomedical insight, critical for clinical epidemiology, leadership in communicable disease control and statutory functions arising from international health threats, where direct clinical involvement was suggested. Clinical roles form part of PH medical specialists in many countries–the US,[44] Canada[45] and Ghana.[46]

Going forward, the policy environment since 2011, with PH units proposed at provincial and district levels together with the NHI, NAPHISA and OHSC, is an opportunity to embed PH graduates and PHM specialists in the services. PH functions need to be identified, their value-add demonstrated, and appropriately located in the services, which are easily dominated by curative and clinical imperatives.

Graduates from postgraduate PH and PHM programmes, should be 'service-ready', able to work at all levels of this reforming health system. Expertise in measurement sciences contributes identifying strengths and gaps in service delivery, particularly at district level. PHM and PH trained physicians were core to municipal health services in pre-apartheid,[47] apartheid [48] and post-apartheid SA, as well as in many countries globally, such as the historical[49] and contemporary British health system,[50] Canada,[51] Italy[52] and Ghana.[53] Specifying PH training for senior district managers, and the speciality for senior professionals managing complex institutions, as in the UK and Norway,[54] requires political will and PHM's convincing, demonstrable value. Roles in health protection, a PH function elsewhere,[22, 38, 55] and in PH agencies, focussed on surveillance and disease control, such as NAPHISA recently promulgated by government,[8] were hardly mentioned.

The divide between academia and the health sector is challenging and solutions must benefit both, as was noted by the Institute of Medicine in the USA, who argued for continuing education for PH service staff and practical placements for students in PH services.[44] New models of academic engagement were welcomed where 'joint' appointment specialists are based in provincial and district services.

Honest discussion between government and PHM stakeholders (universities, CPHM and PHASA) is required to embed the speciality in the services. Conversations should identify PH gaps in the health sector; core competencies for PH graduates and PHM; appropriate training placements for PHM; and service roles suited to the wide range of PH professionals. Then, posts for specific PH professionals could be designated and positions for PHM specialists, created. As junior specialists may first need to work as managers to gain experience, not all PHM specialists working in the state sector would work in designated specialist posts.

**Limitations.** This was a qualitative study and the findings do not intend to be representative of all stakeholders' perspectives on the PHM speciality and specialists' roles. The largely positive perspective about the role of PHM may reflect the high proportion of informants (71%) who were themselves PHM specialists. This was a consequence of their important role in PH and PHM training. Further, this study, conducted in 2012–13, does not fully capture a

number of recent health policy shifts. For example, in 2015, a National Institute of Public Health in South Africa (NAPHISA) was proposed and legislation for its formation was adopted in 2017.[8] Informants perspectives on this institution, which is an important possible employer, could therefore not be sought. Lastly, although senior managers from the private and NGO sectors were interviewed, perceptions about roles in the private sector and NGOs supporting the national Department of Health were not explored, as these informants were historically steeped in SA public sector health policy.

## Conclusions

This study is the first from a LMIC and from Africa to offer insight about Public Health Medicine, the PH medical speciality in South Africa and contributes to international discussions[56] about roles of the speciality. Elsewhere PHM has a firm footprint in public sector health services, but in SA it is not embedded in posts in the health system at any level–municipal, district, provincial or national, save for a few exceptions, where their creation was driven by key individuals. Whilst PHM is not institutionalised in SA, it is acknowledged to be a key discipline for current health reform initiatives in SA. PHM has shifted over time in response to, and in concert with, SA's evolving health system with specialists working in the public sector health system, in a range of roles applying a broad set of competencies. As informants believed, PHM could provide the 'public health intelligence' function, drawing on information derived from routine data, disease and demographic surveillance and international literature to bear on decision-making and policy. Rivalry for posts and employment between PHM and MPH graduates was hardly raised. Rather, they both cadres had flaws, yet both had value and could complement one another.

In particular, PHM was seen as potentially filling a major gap in the health services for a competent cadre of senior health managers, who combine PH and management skills and understand the resource allocation required for appropriate and efficient health services. PHM specialists could contribute at this strategic leadership level, managing complexity but this is unlikely to be the entry point for new specialists or the career destination of all.

Registrar training programmes need to be reviewed to ensure graduates are service-ready, along with solving the institutional challenges to academic PHM. Appropriately located, academic expertise could inform service priority-setting through harnessing PH 'intelligence', whilst, in turn, the services could provide a rich platform for health professional training and research, to mutual benefit.

Issues related to internal factors such as 'identity' and 'culture' of PHM need to be addressed in order to ensure its future success. PHM has a dual 'identity', being a medical speciality and yet part of a multi-disciplinary PH workforce. There are 'cultural' dualities in its scope of practice: it's practice is context dependent/specific yet it has a broad but core disciplinary base. Dualities also exist in its competencies, which are mostly technical, yet not positivist, as it operates firmly within a socially committed ethos. It also needs to advocate both for itself and others in PH in order to have impact.

The changing policy context in South Africa offers the opportunity for structures with PH functions to be institutionalised at all levels in the health services, with posts for PHM specialists and PH professionals. This should facilitate the design and implementation of equitable and accessible health services that incorporate protection, promotion and prevention in addition to curative services in SA's reforming health system.

## Supporting information

**S1 Table. Key informants interviewed.**
(DOCX)

**S1 File. Interview guide.**
(DOCX)

## Acknowledgments

This article is based on VEMZ's doctoral work completed at the University of Cape Town. The authors thank the informants interviewed in this study for their time and insights

## Author Contributions

**Conceptualization:** Virginia E. M. Zweigenthal, Leslie London.

**Data curation:** Virginia E. M. Zweigenthal.

**Formal analysis:** Virginia E. M. Zweigenthal.

**Funding acquisition:** Virginia E. M. Zweigenthal.

**Investigation:** Virginia E. M. Zweigenthal.

**Methodology:** Virginia E. M. Zweigenthal.

**Project administration:** Virginia E. M. Zweigenthal.

**Resources:** Virginia E. M. Zweigenthal.

**Supervision:** William M. Pick, Leslie London.

**Validation:** Virginia E. M. Zweigenthal.

**Visualization:** Virginia E. M. Zweigenthal.

**Writing – original draft:** Virginia E. M. Zweigenthal.

**Writing – review & editing:** Virginia E. M. Zweigenthal, William M. Pick, Leslie London.

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
