## [Decision Letter · Decision Letter 0]

7 Aug 2019

Stakeholders’ perspectives on Public Health Medicine in South Africa

PONE-D-19-15386

Dear Dr. Zweigenthal,

We are pleased to inform you that your manuscript has been judged scientifically suitable for publication and will be formally accepted for publication once it complies with all outstanding technical requirements.

Shortly after the formal acceptance letter is sent, an invoice for payment will follow. To ensure an efficient production and billing process, please log <gwmw class="ginger-module-highlighter-mistake-type-1" id="gwmw-15644694818984069885973">into</gwmw> Editorial Manager at https://www.editorialmanager.com/pone/, click the "Update My Information" link at the top of the page, and update your user information. If you have any billing related questions, please contact our Author Billing department directly at authorbilling@plos.org.

With kind regards,

Kahabi Ganka Isangula,PhD

Academic Editor

PLOS ONE

Additional Editor Comments (optional):

Given my qualitative background, I personally reviewed the manuscript and found it to be well written and scientifically sound. I understand the challenges of reporting qualitative research where, for instance, the authors presented participants' demographic information within the methods sections. I also understand the potential bias that may emerge in selecting themes when an author shares professional characteristics with research participants. However, this is also advantageous in terms of reflexivity. All in all, the paper qualifies for the acceptance decision in line with reviewers' comments. 

Reviewers' comments:

Reviewer's Responses to Questions

**Comments to the Author**

1. Is the manuscript technically sound, and do the data support the conclusions?

Reviewer #1: Yes

Reviewer #2: Yes

2. Has the statistical analysis been performed appropriately and rigorously? 

Reviewer #1: N/A

Reviewer #2: N/A

3. Have the authors made all data underlying the findings in their manuscript fully available?

The PLOS Data policy requires authors to make all data underlying the findings described in their manuscript fully available without restriction, with rare exception (please refer to the Data Availability Statement in the manuscript PDF file). The data should be provided as part of the manuscript or its supporting information, or deposited <gwmw class="ginger-module-highlighter-mistake-type-3" id="gwmw-15644694986901461671757">to</gwmw> a public repository. For example, in addition to summary statistics, the data points behind means, medians and variance measures should be available. If there are restrictions on publicly sharing data—e<gwmw class="ginger-module-highlighter-mistake-type-3" id="gwmw-15644695006329135459744">.</gwmw>g. <gwmw class="ginger-module-highlighter-mistake-type-1" id="gwmw-15644695012553840140252">participant</gwmw> privacy or use of data from a third party—those must be specified.

Reviewer #1: Yes

Reviewer #2: Yes

4. Is the manuscript presented in an intelligible fashion and written in standard English?

PLOS ONE does not copyedit accepted manuscripts, so the language in <gwmw class="ginger-module-highlighter-mistake-type-3" id="gwmw-15644695038473909372248">submitted</gwmw> articles must be clear, correct, and unambiguous. Any typographical or grammatical errors should be corrected at revision, so please note any specific errors here.

Reviewer #1: Yes

Reviewer #2: Yes

5. Review Comments to the Author

Please use the space provided to explain your answers to the questions above. You may also include additional comments for the author, including concerns about dual publication, research ethics, or publication ethics. <gwmw class="ginger-module-highlighter-mistake-type-3" id="gwmw-15644695080623006831576">(</gwmw>Please upload your review as an attachment if it exceeds 20,000 characters)

Reviewer #1: It is a thorough review concerning the Public Health professionals and Public Health Medicine specialist.

Reviewer #2: This is an interesting paper. It addresses a <gwmw class="ginger-module-highlighter-mistake-type-2" id="gwmw-15644695105668431967788">generic</gwmw> problem, but in the context of an African intermediate income country.

6. PLOS authors have the option to publish the peer review history of their article (what does this mean?). If published, this will include your full peer review and any attached files.

Reviewer #1: No

Reviewer #2: No

<gdiv></gdiv>

---

## [Editor Report · Acceptance letter]

16 Aug 2019

PONE-D-19-15386 

Stakeholders’ perspectives on Public Health Medicine in South Africa 

Dear Dr. Zweigenthal:

I am pleased to inform you that your manuscript has been deemed suitable for publication in PLOS ONE. Congratulations! Your manuscript is now with our production department. 

With kind regards,

on behalf of

Dr. Kahabi Ganka Isangula 

Academic Editor

PLOS ONE